# Implementation and evaluation of the unified stomatal optimization approach in the Functionally Assembled Terrestrial Ecosystem Simulator (FATES)

Qianyu Li[1], Shawn P. Serbin[1], Julien Lamour[1], Kenneth J. Davidson[1,2], Kim S. Ely[1], Alistair Rogers[1]

[1]Department of Environmental and Climate Sciences, Brookhaven National Laboratory, Upton, NY

[2] Department of Ecology and Evolution, Stony Brook University, Stony Brook, NY

*Correspondence to*: Qianyu Li (qli1@bnl.gov)

**Abstract.** Stomata play a central role in regulating the exchange of carbon dioxide and water vapor between ecosystems and the atmosphere. Their function is represented in land surface models (LSMs) by conductance models. The Functionally Assembled Terrestrial Ecosystem Simulator (FATES) is a dynamic vegetation demography model that can simulate both detailed plant demographic and physiological dynamics. To evaluate the effect of stomatal conductance model formulation on forest water and carbon fluxes in FATES, we implemented an optimality-based stomatal conductance model—the Medlyn (MED) model, that simulates the relationship between photosynthesis ($A$) and stomatal conductance to water vapor ($g_{sw}$) as an alternative to the FATES default Ball-Woodrow-Berry (BWB) model. To evaluate how the behavior of FATES is affected by stomatal model choice, we conducted a model sensitivity analysis to explore the response of $g_{sw}$ to climate forcing, including atmospheric $CO_2$ concentration, air temperature, radiation, and vapor pressure deficit in the air ($VPD_a$). We found that modeled $g_{sw}$ values varied greatly between the BWB and MED formulations due to the different default stomatal slope parameters ($g_1$). After harmonizing $g_1$ and holding the stomatal intercept parameter ($g_0$) constant for both model formulations, we found that the divergence in modeled $g_{sw}$ was limited to conditions when the $VPD_a$ exceeded 1.5 kPa. We then evaluated model simulation results against measurements from a wet evergreen forest in Panama. Results showed that both the MED and BWB model formulations were able to capture the magnitude and diurnal changes of measured $g_{sw}$ and $A$ but underestimated both by about 30% when the soil was predicted to be very dry. Comparison of modeled soil water content from FATES to a reanalysis product showed that FATES captured soil drying well but translation of drying soil to modeled physiology reduced the models' ability to match observations. Our study suggests that the parameterization of stomatal conductance models and current model response to drought are the critical areas for improving model simulation of $CO_2$ and water fluxes in tropical forests.

**Keywords:** FATES, stomatal conductance, photosynthesis, optimization theory, soil moisture, tropical forests

## 1 Introduction

Global climate change has resulted in significant modifications to Earth's ecosystems through changing weather patterns, including an increased frequency and severity of extreme drought and heatwaves, which has resulted in

increased risk for terrestrial vegetation (Pachauri et al., 2014; Reichstein et al., 2013; Gatti et al., 2021). The exchange of water vapor and carbon dioxide between plants and the atmosphere is dominated by transport through stomata (Hetherington and Woodward, 2003; Kala et al., 2016). The mechanisms regulating stomatal opening involve complex biochemical and biophysical processes that are currently not represented in land surface models (LSMs) (Lawson et al., 2014; Buckley and Mott, 2013; Blatt, 2000; Davies et al., 2002). However, a range of much simpler, largely empirical, formulations that describe the responses of stomata to their environment have been successfully used by LSMs for many years. Most of them require only two parameters i.e. the intercept parameter ($g_0$), which is the conductance when photosynthesis ($A$) is zero, and the slope parameter ($g_1$) that describes the relationship between stomatal conductance to water vapor ($g_{sw}$) and a regressor term that includes $A$ and environmental drivers (Damour et al., 2010; Berry et al., 2010). The most widely used representation of $g_{sw}$, and the default formulation used in the Functionally Assembled Terrestrial Ecosystem Simulator (FATES), is the Ball-Woodrow-Berry model (BWB, Ball et al., 1987), where $g_{sw}$ is based on an empirical relationship with leaf net photosynthesis ($A_{net}$), carbon dioxide concentration at the leaf surface ($C_s$), and relative humidity at the leaf surface ($H_s$) as Eq. (1):

$$g_{sw} = g_0 + g_1 \frac{A_{net}}{C_s} H_s, \tag{1}$$

The optimality-based unified stomatal conductance model (Medlyn model, MED, Medlyn et al., 2011) is based on the assumption that plants will attempt to maximize carbon gain while minimizing water loss (Cowan and Farquhar, 1977). The MED model has been proposed as an alternative representation of $g_{sw}$ in LSMs (De Kauwe et al., 2015; Lawrence et al., 2019). The basic functional form of the MED model is shown in Eq. (2). One important difference between the BWB and MED formulations is that $g_{sw}$ responds to vapor pressure deficit at the leaf surface ($VPD_s$) instead of $H_s$:

$$g_{sw} = g_0 + 1.6\left(1 + \frac{g_1}{\sqrt{VPD_s}}\right)\frac{A_{net}}{C_s}, \tag{2}$$

Although the functional form of the MED model is similar to the BWB model, $g_1$ is based on underlying optimization theory and has a strong theoretical link to plant water use efficiency. The MED model is also favored by many plant physiologists given that $g_{sw}$ responds $VPD_s$ rather than $H_s$ (Rogers et al., 2017a). Importantly, the $g_1$ parameter has also been found to vary significantly across a wide range of different plant functional types (PFTs) and climate regions (Lin et al., 2015). Better representation of $g_{sw}$ in LSMs requires efforts to improve the fidelity of $g_1$ parametrization by PFT. The $g_1$ parameter can be estimated through field measurement campaigns (Lin et al., 2015; Wu et al., 2020) or model inversion (Bonan et al., 2011; Fer et al., 2018). For example, De Kauwe et al. (2015) derived a PFT-specific $g_1$ parameterization from Lin et al. (2015) for the CABLE model and found a significant reduction in annual fluxes of transpiration using MED compared with the original model formulation of CABLE (Leuning, 1995). Despite considerable analysis supporting the adoption of the MED model in LSMs (De Kauwe et al., 2014; Lawrence et al., 2019), the formulation has not been widely adopted and is not common in dynamic vegetation models (Fisher et al., 2018).

Exploring plant physiological responses to key environmental variables is emerging as a promising way to understand model representation and evaluate model behaviors (Rogers et al., 2017a; Bonan et al., 2011). Stomatal conductance in the MED and BWB models responds to direct environmental drivers including atmospheric $CO_2$ concentration and $VPD_s$ or $H_s$, and indirect drivers like radiation and leaf surface temperature (via photosynthesis) (Franks et al., 2017). Evaluating the BWB and MED formulations to changing climate in a complete LSM, where atmospheric, ecological, and hydrologic processes are highly coupled, is urgently needed to understand model responses within a larger domain.

Improved projection of the response of ecosystems to global climate change requires an improved understanding and model representations of plant responses to a hotter, drier, and $CO_2$ enriched future (Sullivan et al., 2020). To mimic the drought effects on ecosystems, some models have included a soil water stress factor (often denoted as $\beta$) which is used to reduce the "base rate" of stomatal model parameters, either $g_0$ (e.g., CLM, Lawrence et al., 2019), $g_1$ (e.g., G'DAY, Comins and McMurtrie, 1993; O-CN, Zaehle and Friend, 2010; CABLE, De Kauwe et al., 2015), or both (e.g., ORCHIDEE, Guimberteau et al., 2018). In some cases, it is also used to lower the maximum carboxylation rate of Rubisco ($V_{cmax}$) (e.g., CLM; O-CN; SIBCASA, Schaefer et al., 2008), both $V_{cmax}$ and the maximum rate of electron transport ($J_{max}$) (e.g., G'DAY), or directly $A$ (e.g., JULES, Best et al., 2011; Clark et al., 2011). Reduction in $A$ will further reduce $g_{sw}$. Some models also consider the soil water stress on mesophyll conductance (e.g., SIBCASA; ORCHIDEE). However, the application of a $\beta$ factor on different physiological parameters has not been evaluated against measurements for $g_{sw}$ models. Therefore, evaluating different $g_{sw}$ schemes and parameterization with data collected under normal and stressed conditions may help reveal areas for model improvement.

In this study, we explored the impact of stomatal behavior under simulated and realistic environmental conditions in the FATES model (Koven et al., 2020), where we implemented the MED formulation as an alternative approach to the default BWB formulation. The FATES model is a dynamic vegetation demography model that simulates leaf to ecosystem-scale carbon, water, and energy fluxes, as well as cohort-level plant growth, competition, and mortality processes, enabling FATES to predict the distribution, structure, and composition of vegetation (Fisher et al., 2015; Koven et al., 2020). FATES itself is not a standalone model, but instead is used in conjunction with a host land model, and is currently coupled with the Community Land Model (CLM, Lawrence et al., 2019) and the Energy Exascale Earth System Model (E3SM) Land Model (ELM, Holm et al., 2020). Using FATES and the MED and BWB representations we addressed the following questions: (1) How do projected leaf-level and canopy-level $CO_2$ and water vapor fluxes differ between the BWB and MED formulations in response to key meteorological forcing variables? (2) How do the two model outputs of stomatal conductance and photosynthesis compare to leaf-level gas exchange measurements collected through a dry season in a tropical forest? (3) How does the application of a soil water stress factor affect the simulation of water and carbon cycles during dry periods in a tropical forest?

## 2 Methods

### 2.1 Implementation of the Medlyn model into FATES

In FATES, leaf-level photosynthesis ($A$) in $C_3$ plants is based on the model of Farquhar et al. (1980) as modified by Collatz et al. (1991). $A$ is calculated as the minimum of RuBP carboxylase (Rubisco) limited rate and RuBP regeneration rate (i.e., the light-limited rate). Net photosynthesis rate ($A_{net}$) is the difference between $A$ and leaf respiration. GPP is calculated as the weighted average of the photosynthetic rate from sunlit and shaded leaves, which is integrated through the vertical profile, and finally across all the leaf layers by multiplying exposed leaf area for a given cohort. A cohort is a group of plants with similar disturbance history, height, and PFT type. The leaf area of each cohort is calculated from leaf biomass and specific leaf area (SLA). Leaf biomass is controlled by the processes of phenology, allocation, and turnover. SLA is a PFT-specific parameter.

We implemented the MED stomatal conductance model as an alternative to the BWB model for the calculation of $g_{sw}$ in FATES. Leaf-level $g_{sw}$ is central to the water, $CO_2$ and energy cycles in forests. It not only controls the water and $CO_2$ exchange, but also modifies the energy balance and biochemical processes. Similarly, in FATES, the variable $g_{sw}$ is used to model several processes such as the heat and water transfer and photosynthesis. The calculation of this variable is therefore complex and uses both analytical and numerical solutions to couple the equations describing each process. A detailed description of the implementation can be found in online documentation (FATES Development Team, 2020b). It should be noted that parameters $g_0$ and $V_{cmax}$ used to calculate $A_{net}$ in Eq. (1) and Eq. (2) are multiplied by an empirical soil moisture stress factor ($\beta$) by default in the FATES model. The $\beta$ factor ranges from one when the soil is wet to zero when the soil is dry. The $\beta$ factor depends on the soil water potential of each soil layer, the root distribution of the PFT, and a plant-dependent response to soil water stress as shown in Eq. (3):

$$\beta = \sum_{j=1}^{nj} w_j r_j, \tag{3}$$

where $w_j$ is a plant wilting factor for layer $j$ and $r_j$ is the fraction of roots in layer $j$. The soil wilting factor is a bounded linear function of soil matric potential, defined by two parameters, the soil water potential at (and above) which stomata are fully open, and the value at which stomata are fully closed. The soil matric potential is related to the soil water content, soil texture, and organic matter content. The root fraction is determined by PFT-specific root distribution parameters. For more details on the calculation of the plant wilting factor and the fraction of roots, see the CLM version 4.5 (CLM4.5) technical note (Oleson et al., 2013).

### 2.2 The San Lorenzo, Panama Model Testbed Site

Our model simulations were made on a single tropical forest located in Bosque Protector San Lorenzo, Panama (9°16′51.71″ N, 79°58′28.27″ W, elevation 25 m), which is a part of the Center for Tropical Forest Science (CTFS) –Forest Global Earth Observatory (ForestGEO). The Smithsonian Tropical Research Institute canopy crane provides access to the top of the forest canopy and allows us to compare our simulations with previous measurements of stomatal conductance and net photosynthesis rate (Wu et al., 2020; Rogers et al., 2017c). The site is characterized as a moist

tropical forest, with mean annual temperature of 26 °C, with only small seasonal variation. The mean annual precipitation is 3300 mm, 90 % of this precipitation falls during the wet season (May-December). More details about the site can be found in Wright et al. (2003).

For our study, we conducted all model simulations using the FATES model coupled with the CLM version 5.0.34 (CLM5). For all simulations, we initialized the FATES model using real-world forest inventory data that provided information on tree size distribution for the whole forest area (Condit et al., 2009) and enabled us to better compare model outputs with field measurements by matching the internal cohort structure with that observed in inventory data. Inventory data from the most recent census (1999) were used as the initial state for the simulations. For simplicity, in our FATES simulations we assumed that the site is populated entirely by the broadleaf evergreen tropical (BET) tree plant functional type.

### 2.3 Sensitivity simulations of FATES with synthetic forcing

The MED and BWB stomatal conductance models differ in the representation of atmospheric dryness as well as the $g_1$ values. To isolate the influence of structural and parametric differences on FATES simulations using the MED and BWB stomatal models, we employed three model ensemble simulations, associated with a BET tree PFT.

For the BWB configuration we used the BWB model with a default $g_1$ value of 8 (unitless) for the BET tree PFT in FATES. In our MED-default setup, the MED model was parameterized with $g_1$ set to 4.1 $kPa^{0.5}$ to match the best estimate from Lin et al. (2015). To constrain the model difference to structural difference we also ran FATES with the MED model with a $g_1$ value that was harmonized with the BWB model in FATES, which was abbreviated as MED-B. Here, we assumed $g_1$ for BWB ($g_{1b}$) = 8, air temperature = 25 °C, and relative humidity in the air ($H_a$) = 0.8 following Franks et al. (2017) in Eq. (4) to obtain a BWB-equivalent $g_1$ = 2.39 $kPa^{0.5}$ in the MED-B simulation ($g_{1m}$, Eq. 4), where $VPD_a$ is $VPD$ in the air. For all simulations, $g_0$ was fixed at 1000 $\mu$mol $m^{-2}$ $s^{-1}$.

$$g_{1b} = \frac{1.6}{H_a}\left(1 + \frac{g_{1m}}{\sqrt{VPD_a}}\right), \tag{4}$$

The FATES model is driven by half-hourly longwave radiation, shortwave radiation, air temperature, specific humidity, precipitation, surface pressure, wind speed, and atmospheric $CO_2$ concentration. These variables modify the leaf conductance by changing the environment at the leaf surface ($H_s$, $VPD_s$, and $C_s$ in Eq. (1) & (2)). Following model initialization, the model was run with our synthetic climate forcing data in order to reveal model responses to specific climate forcing. Our synthetic climate forcing, each represented the scenarios of a linear increase in $VPD_a$, air temperature ($T_a$), photosynthetically active radiation (PAR), and atmospheric $CO_2$ concentration ($CO_2$), respectively, while other climate forcing data were kept as constant. The details for these scenarios are listed in Table 2. In addition, we set the precipitation to 1.3 mm $day^{-1}$, surface pressure to 99626 Pa, wind speed to 4.8 m $s^{-1}$, and longwave radiation to 407.4 W $m^{-2}$ for all these scenarios, which represent annual average conditions at our field site. Given the physical dependence of saturated water vapor on $T_a$ (Ficklin and Novick, 2017), it was necessary to adjust the specific humidity together with $T_a$ to keep the $VPD_a$ fixed at 1 kPa for the MED-default and MED-B simulations. For the $T_a$ scenario in

the BWB model, $H_a$ was kept as 80 % as $H_a$ rather than $VPD_a$ was used in the BWB model (Franks et al., 2017). We then studied the responses of $g_{sw}$, net photosynthesis ($A_{net}$), gross primary productivity (GPP), and evapotranspiration (ET) to these drivers for the top layer (averaged across sunlit and shaded leaves) of the canopy. We also checked the number of plants per hectare (nplant) to ensure that cohort density did not change during our simulations.

Table 2. Scenario setting for the sensitivity simulations

| Scenario | $VPD_a$ (kPa) | $T_a$ (°C) | PAR ($\mu$mol m$^{-2}$ s$^{-1}$) | $CO_2$ (ppm) |
|---|---|---|---|---|
| $VPD_a$ | 0-2.5 | 25 | 1500 | 400 |
| $T_a$ | 1 | 5-50 | 1500 | 400 |
| Radiation | 1 | 25 | 0-2000 | 400 |
| $CO_2$ | 1 | 25 | 1500 | 100-1000 |

**2.4 Evaluation of FATES against in situ measurements**

We compared the modeled diurnal $g_{sw}$ and $A_{net}$ of upper canopy layers with measured values (Wu et al., 2020). The data were collected at the San Lorenzo field site at monthly intervals across the dry season and the beginning of the wet season during a strong ENSO year 2016. The $g_{sw}$ and $A_{net}$ of top of canopy leaves of eight species which all belonged to BET PFT were measured across four months starting in February 2016 and ending in May 2016. Measurements of $g_{sw}$ and $A_{net}$ were made with a LI-6400 (LiCor Biosciences, Lincoln, NE, USA) where the conditions of radiation, humidity, $CO_2$ concentration, and temperature surrounding the leaves were closely matched to the ambient conditions. Using this dataset, the $g_1$ values of the BWB and MED models were estimated for each species (see Table 2 in Wu et al., 2020). We used those estimations to parametrize the $g_{sw}$ model in FATES and we used their $g_{sw}$ and $A_{net}$ measurements to compare with FATES simulation results. It should be noted that the $g_1$ values we used were varied between 4.43 to 8.3 for the BWB model, and between 1.14 kPa$^{0.5}$ and 2.85 kPa$^{0.5}$ for the MED model, most values were lower than the defaults for evergreen tropical trees in both of the models, as discussed in Wu et al. (2020). Because $g_1$ was estimated for BWB and MED models based on the same measurements, $g_1$ was equivalent for the two models and the simulations resemble MED-B and BWB in section 2.3. An ensemble of simulations with varying measured species-specific $g_1$ values were carried out to evaluate the impact of stomatal slope parameterization on FATES simulated $g_{sw}$ and $A_{net}$. In addition, $V_{cmax}$ at 25 °C was set to 63 $\mu$mol m$^{-2}$ s$^{-1}$ based on the $A/Ci$ curves measured at the same time during the 2016 campaign (Rogers et al., 2017b). Other parameters such as $J_{max}$ and leaf dark respiration rate ($R_{dark}$) at 25 °C were directly calculated by FATES based on their relationship with $V_{cmax}$ or leaf nitrogen content.

For our simulations, we used the observed half-hourly weather data including precipitation, air temperature, and humidity from the field site meteorological station as the atmospheric forcing data to drive the FATES simulations (Faybishenko et al., 2019). Atmospheric $CO_2$ concentration was set to a background level of 403.3 ppm based on data

from the NOAA's Mauna Loa observatory, which is also very close to the $CO_2$ concentration inside the leaf chamber of the gas exchange equipment.

## 2.5 Drought effects on physiological parameters

In FATES, a soil water stress factor ($\beta$) is used to adjust $g_0$ and $V_{cmax}$ in the original form of the BWB model (Bonan et al., 2011). For the MED approach we implemented, we also applied the $\beta$ factor in the same manner as the default

setting (Sect. 2.1). However, whether the calculation of the $\beta$ factor can truly reflect soil water conditions is unclear. To the best of our knowledge, the relevance of the $\beta$ factor has not been rigorously tested for tropical ecosystems, in comparison with measured $g_{sw}$ and $A_{net}$, either. We therefore first compared the modeled soil water content and $\beta$ factor against soil moisture products of ECMWF Reanalysis data version 5 (ERA5) (Hersbach et al., 2018). Then we explored whether this formulation of the $\beta$ factor accurately represents observed physiological responses to soil water

stress, and whether the stress factor should also be applied to $g_1$ for both the BWB and MED models. To test this, we designed model simulations (Table 3) to assess how the inclusion of the $\beta$ factor modifies modeled $g_{sw}$ and $A_{net}$ and the comparison with the measurements. In these simulations, $g_1$ and $V_{cmax}$ were set as the averages across all the species measured for the BET PFT.

Table 3. Model simulations for studying soil water stress effects on physiological parameters in FATES

| Experiment | $g_0$ | $g_1$ | $V_{cmax}$ |
|---|---|---|---|
| Default | on | off | on |
| Exp 1 | on | off | off |
| Exp 2 | on | on | on |
| Exp 3 | on | on | off |
| Exp 4 | off | off | off |

on = soil water stress effect is turned on, off = soil water stress effect is turned off in the simulation.

## 3 Results

### 3.1 Model responses to climatic drivers

The responses of $g_{sw}$ and $A_{net}$ to climatic forcing as modeled by the BWB (BWB model with default $g_1$) and MED-default (MED model with default $g_1$) simulations had similar shapes (Fig. 1&2, blue and black lines). The MED-default yielded markedly higher $g_{sw}$ than BWB for all climatic drivers considered with an average difference of 75 % (Fig. 1). The MED-default simulation also resulted in higher estimations of $A_{net}$ but the increase over the BWB simulation was much smaller (around 15 % on average) than the increase in $g_{sw}$ (Fig. 2). When the $VPD_a$ increased

above 1.5 kPa, the two models showed a strong additional divergence. At a $VPD_a$ of 2.0 kPa projected $g_{sw}$ and $A_{net}$ from the BWB simulation were 316 % and 86 % lower than projections from the MED-default simulation (Fig. 1c&2c).

The particularly large divergence between the BWB and the MED-default simulations can be explained by a combination of parametric and structural differences. Comparison of the MED-B (where we used a parameterization equivalent to that of BWB in the MED model) with the BWB limited potential model deviation to structural difference between the two approaches. Both simulations yielded similar responses of $g_{sw}$ and $A_{net}$ to radiation, temperature, and $CO_2$ (Fig.1a,1b,1d, 2a, 2b&2d, blue and red lines), demonstrating that the differences between the BWB and MED-default settings were attributable to the difference in parameterization associated with $g_1$. With a harmonized parameterization of $g_1$ the divergence between the two models above a $VPD_a$ of 1.5 kPa was still readily apparent (Fig. 1c&2c, blue and red lines). The MED-B simulation showed a slight decrease of $g_{sw}$ with high $VPD_a$ while $g_{sw}$ modeled with the BWB simulation decreased more markedly when $VPD_a$ was beyond 1.5 kPa. At 2.0 kPa the BWB simulation projected $g_{sw}$ and $A_{net}$ that were 126 % and 53 % lower than the MED-B simulation. For the temperature response of $g_{sw}$, BWB and MED-B were very similar although BWB had slightly higher $g_{sw}$ values than MED-B (Fig. 1d, blue and red lines).

In contrast to the $g_{sw}$ response, the differences between BWB and MED-default were generally smaller for $A_{net}$, except when $VPD_a$ was above 1.5 kPa (Fig. 2, blue and black lines). The use of measured $g_1$ for the MED model (MED-default) did not markedly change the magnitude of $A_{net}$ compared with MED-B (Fig. 2, red and black lines). When we explored the ecosystem-scale responses (Fig. 3&4), we found that the patterns of ET and GPP mirrored the leaf-level responses described above when using our synthetic climatic drivers. The difference between BWB and MED-B was also apparent when $VPD_a$ was above 1.5 kPa (Fig. 3c&4c, blue and red lines).

To rule out that these differences were related to changes in underlying plant community structure, we looked for any significant changes in cohort density (number of plants per hectare). Our results showed that there was no significant change (Fig. S1) thus these ecosystem-scale responses were primarily related to changes in underlying leaf-level physiology.

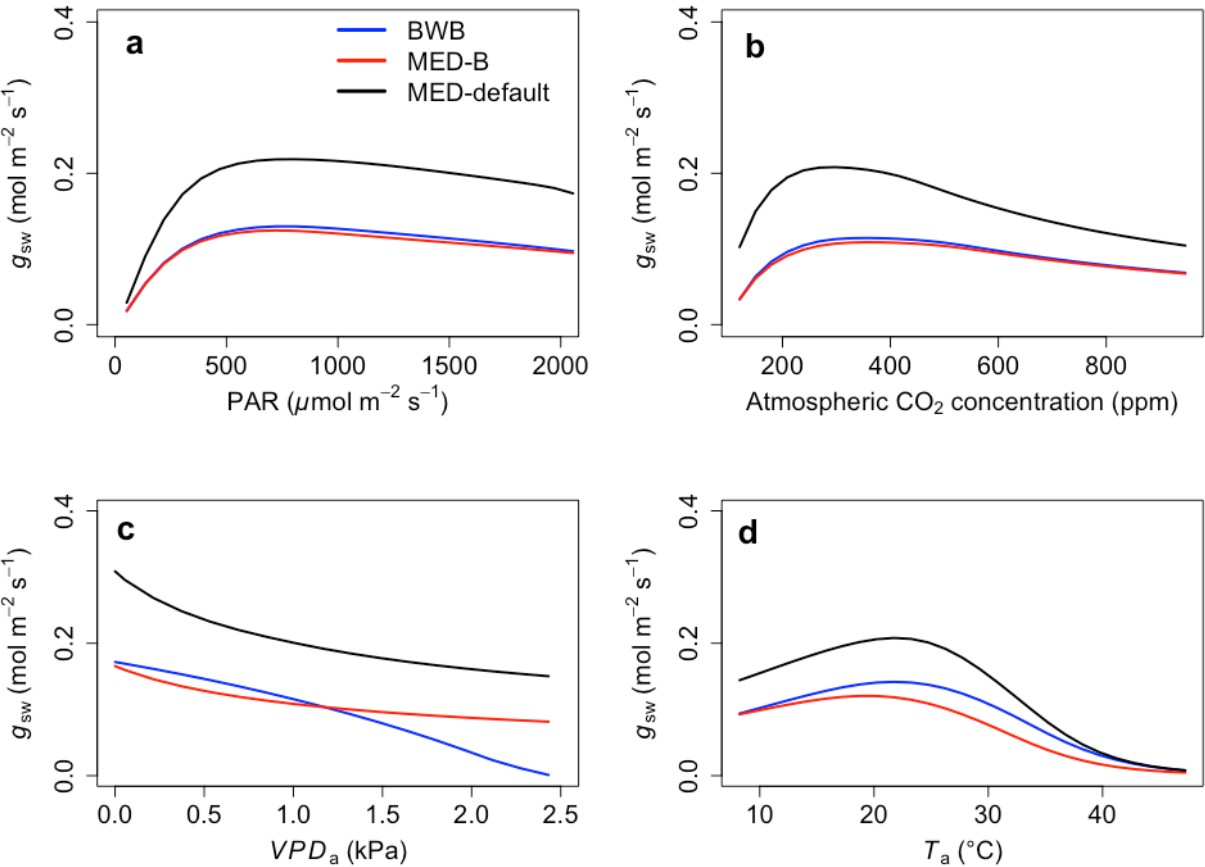

**Fig. 1. The responses of stomatal conductance ($g_{sw}$) to scenarios (a) Radiation, (b) $CO_2$, (c) $VPD_a$, and (d) $T_a$ for the three model setups: BWB (blue), MED-B (red), and MED-default (black).**

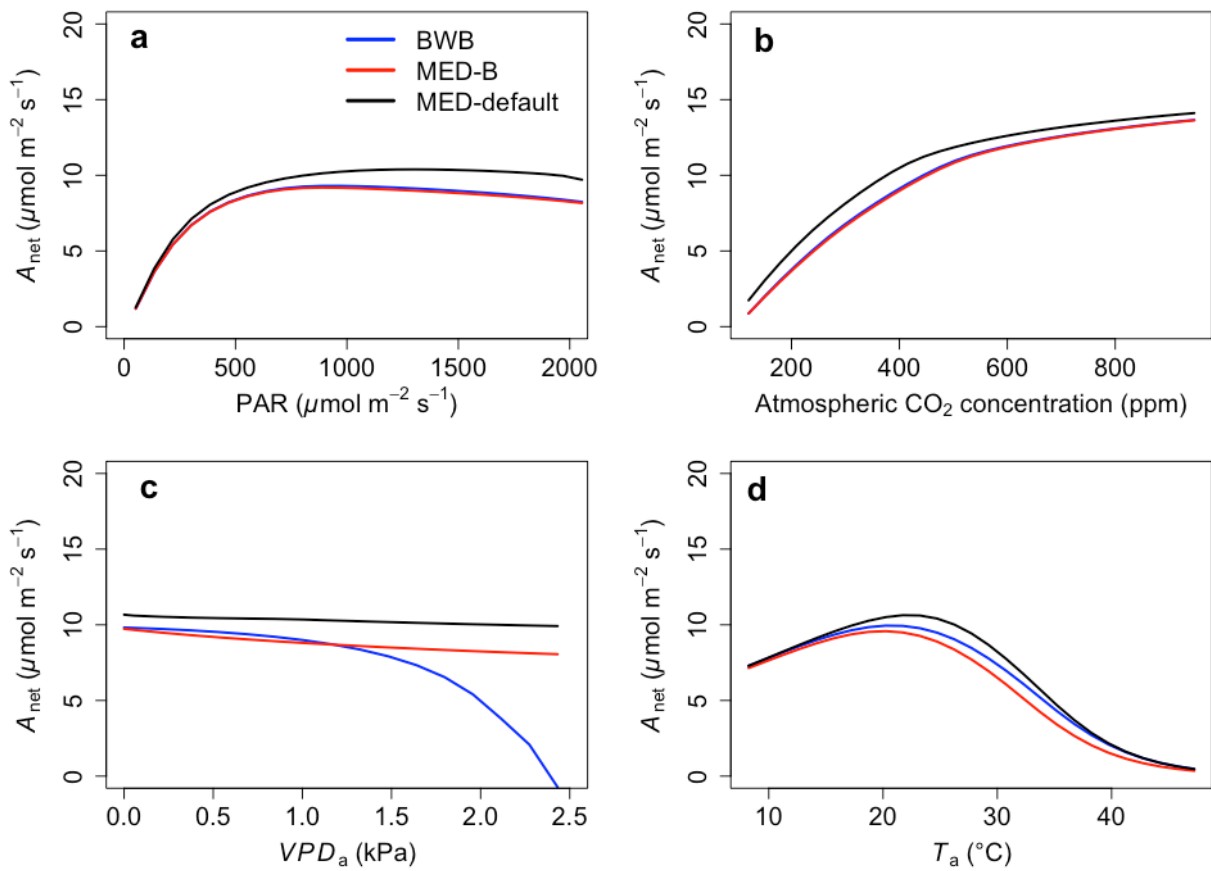

**Fig. 2. The responses of net photosynthesis ($A_{net}$) to scenarios (a) Radiation, (b) $CO_2$, (c) $VPD_a$, and (d) $T_a$ for the three model setups: BWB (blue), MED-B (red), and MED-default (black).**

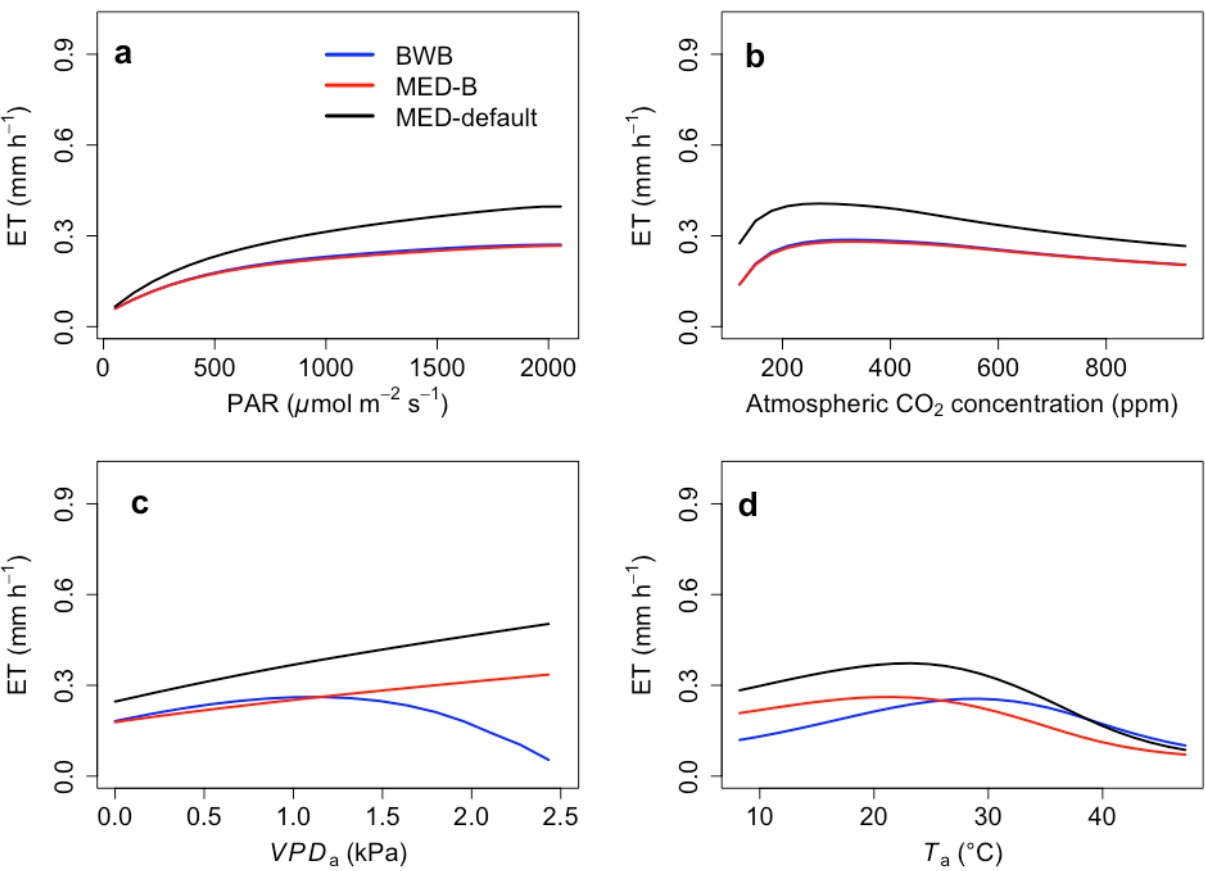

Fig. 3. The responses of evapotranspiration (ET) to scenarios (a) Radiation, (b) $CO_2$, (c) $VPD_a$, and (d) $T_a$ for the three model setups: BWB (blue), MED-B (red), and MED-default (black).

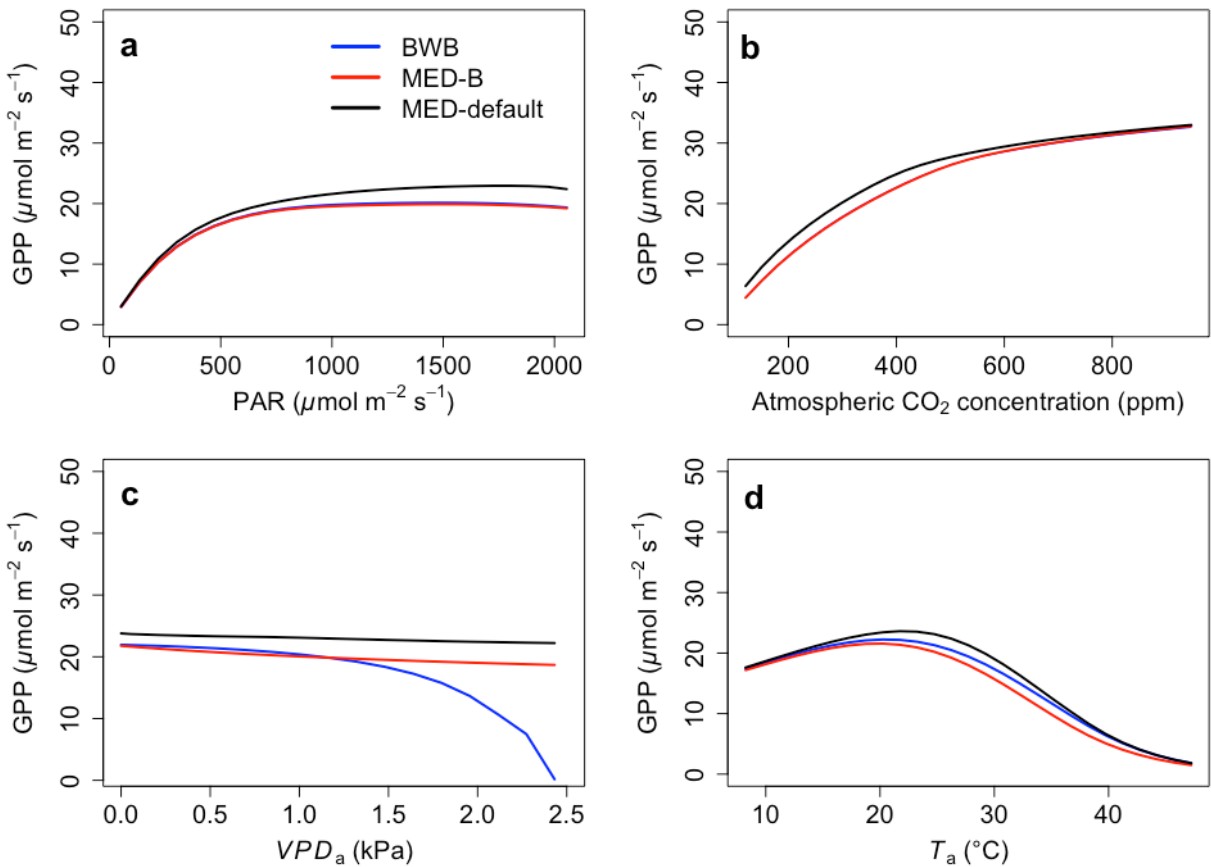

Fig. 4. The responses of gross primary productivity (GPP) to scenarios (a) Radiation, (b) $CO_2$, (c) $VPD_a$, and (d) $T_a$ for the three model setups: BWB (blue), MED-B (red), and MED-default (black).

### 3.2 Model evaluation against field measurements

Before comparing the results of the BWB and MED model representations within FATES against field measurements, we first evaluated the consistency of the site meteorological measurements used to drive FATES simulations with those measured with our gas exchange instruments during the campaign. We anticipated that the two conditions would be comparable as the environmental controls in the gas exchange instruments were set to mimic the ambient conditions just before the leaf measurements. We found that for PAR, $H_a$, and $CO_2$ concentration, the atmospheric and leaf chamber conditions at time of measurements were in reasonably close agreement, while the in situ measured $T_a$ and $VPD_a$ were higher than climate data in all months (Fig. 5a-5d).

To account for measurement and natural variability of $g_1$ across different species, we ran a series of FATES simulations driven by meteorological forcing data with different $g_1$ values. These experiments showed that FATES $A_{net}$ and $g_{sw}$ were sensitive to different $g_1$ values for both model formulations (Fig. 5e-5l). The MED model ensemble results of $A_{net}$ and $g_{sw}$ with different $g_1$ values, represented as the envelopes in Fig. 5e-5l, generally overlapped with those from the BWB model, with comparable averages. Compared with field measurements, both models captured

the diurnal patterns well (Fig. S2) but tended to underestimate $A_{net}$ and $g_{sw}$ notably in the month of April by about 30 %, at the peak of the dry season (Fig. 5g&5k).

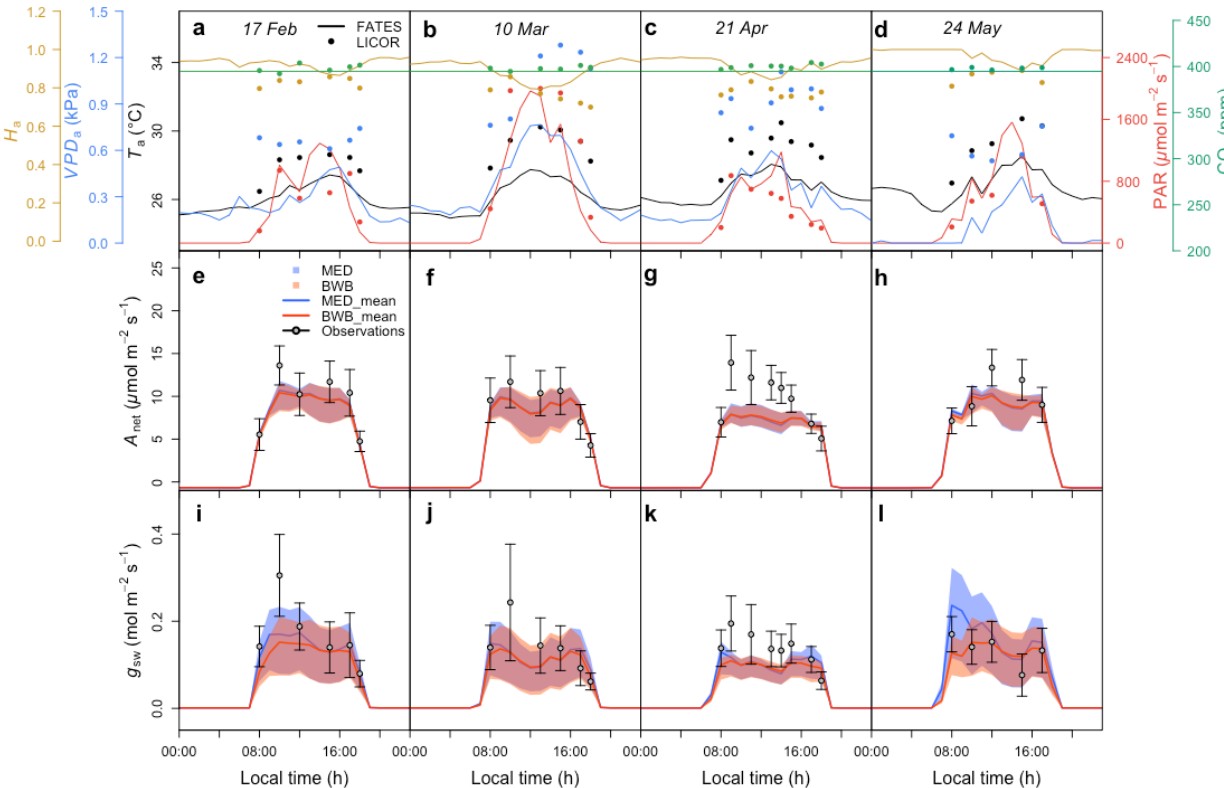

**Fig. 5. (a-d) Diurnal change in climate forcing, (e-h) model-data comparison of net photosynthesis rate ($A_{net}$), (i-l) model-data comparison of stomatal conductance ($g_{sw}$) for four field campaign dates. In panel (a-d), lines and filled points represent climate forcing data used in FATES and in situ measurements, respectively. Different colors are for different types, black for $T_a$, red for PAR, blue for $VPD_a$, green for atmospheric $CO_2$ concentration, and gold for $H_a$. In panel (e-l) shading areas represent range of FATES model ensemble results with different measured $g_1$ values for different species, while lines represent the averages of these ensemble results. Blue shading areas and lines are for results from the MED model, and red for the BWB model. Gray filled circles for the measured data represent averages across species. Black error bars for the measured data represent the 95 % CI across species. Columns correspond to days of measurements and are presented in chronological order for 17 February, 10 March, 21 April, and 24 May in 2016.**

### 3.3 Water stress factor on physiological parameters

Compared with the ERA5 soil moisture products, FATES generally captured the magnitude and trend of the observed average soil water content at the San Lorenzo site (Fig. 6). FATES also simulated the soil water content well for different layers of the soil column (Fig. S3). By April 2016, at the peak of the dry season in a dry year, the simulated soil moisture stress factor (averaged over all the soil layers) reached an annual minimum (0.7), corresponding to the observed soil moisture drying trend (Fig. 6). FATES also underestimated $g_{sw}$ and $A_{net}$ by the largest margin in April, when compared to our field measurements (Fig. 5g&5k). To explore this further, we conducted additional experiments focused on evaluating the use of the $\beta$ factor to modify $g_0$, $g_1$, and $V_{cmax}$. For the month of April in 2016, we compared a range of different model simulation experiments where the $\beta$ factor was applied in different combinations to $g_0$, $g_1$, and $V_{cmax}$ (Table 3, Fig. 7). The results from Exp. 1 and Exp. 4 showed high overlap, indicating that considering the $\beta$ effect on $g_0$ does not influence modeled carbon and water fluxes. However, when applied to $V_{cmax}$ the $\beta$ factor reduced

$g_{sw}$ and $A_{net}$ by 15 %-20 % (Exp. 2 vs. Exp. 3, Fig. 7). Applying the $\beta$ factor to $g_1$ also reduced $g_{sw}$ and $A_{net}$ by 10 %-50 % (Exp. 1 vs. Exp. 3, Fig. 7). Unsurprisingly, comparing model results with $\beta$ applied to all or no parameters showed the largest differences (30 %-80 %) (Exp. 2 vs. Exp. 4, Fig. 7). Default simulations with the $\beta$ factor on $g_0$ and $V_{cmax}$ underestimated $A_{net}$ by 29 %, and $g_{sw}$ by 26 % for the MED model. However, the results from simulations with no $\beta$ effects or with $\beta$ only applied to $g_0$ (Exp. 1&4) corresponded best to the observations, in which $A_{net}$ was only

underestimated by 15 %, and $g_{sw}$ by 9 % for the MED model (Fig. 7a&7c). There was also a significant improvement of performance when the $\beta$ effects were removed from equation in the BWB model (Fig. 7b&7d).

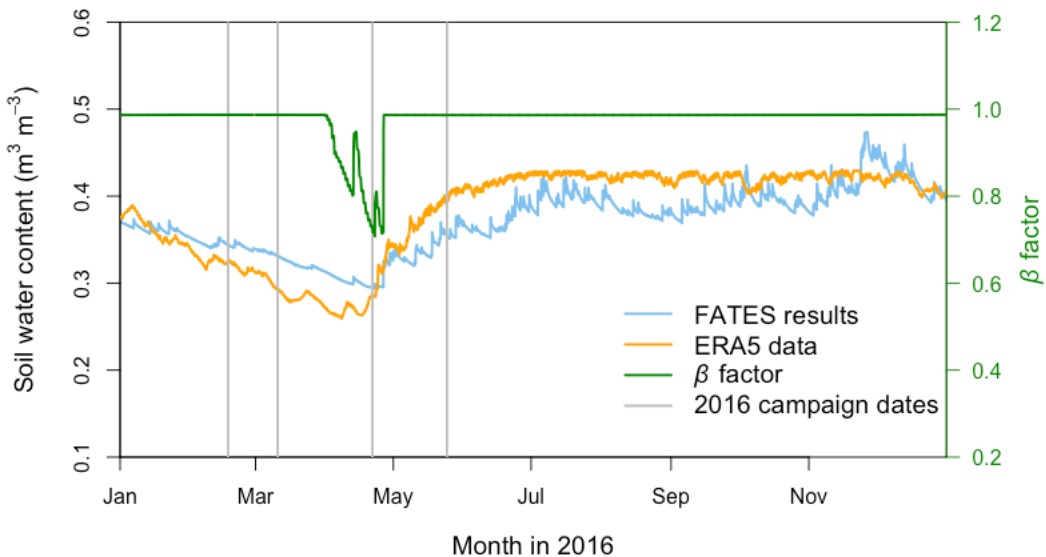

**Fig. 6. Annual cycle of modeled volumetric soil water content (blue line) and corresponding soil water stress ($\beta$) factor (green line) from FATES simulation, and ERA5 reanalysis soil water content data (orange line) at the San Lorenzo field**

**site in 2016. The soil water content data are means across all soil layers. For the $\beta$ factor, "1" represents fully saturated soil, while "0" represents very dry soil. Vertical gray lines indicate the four campaign dates in 2016.**

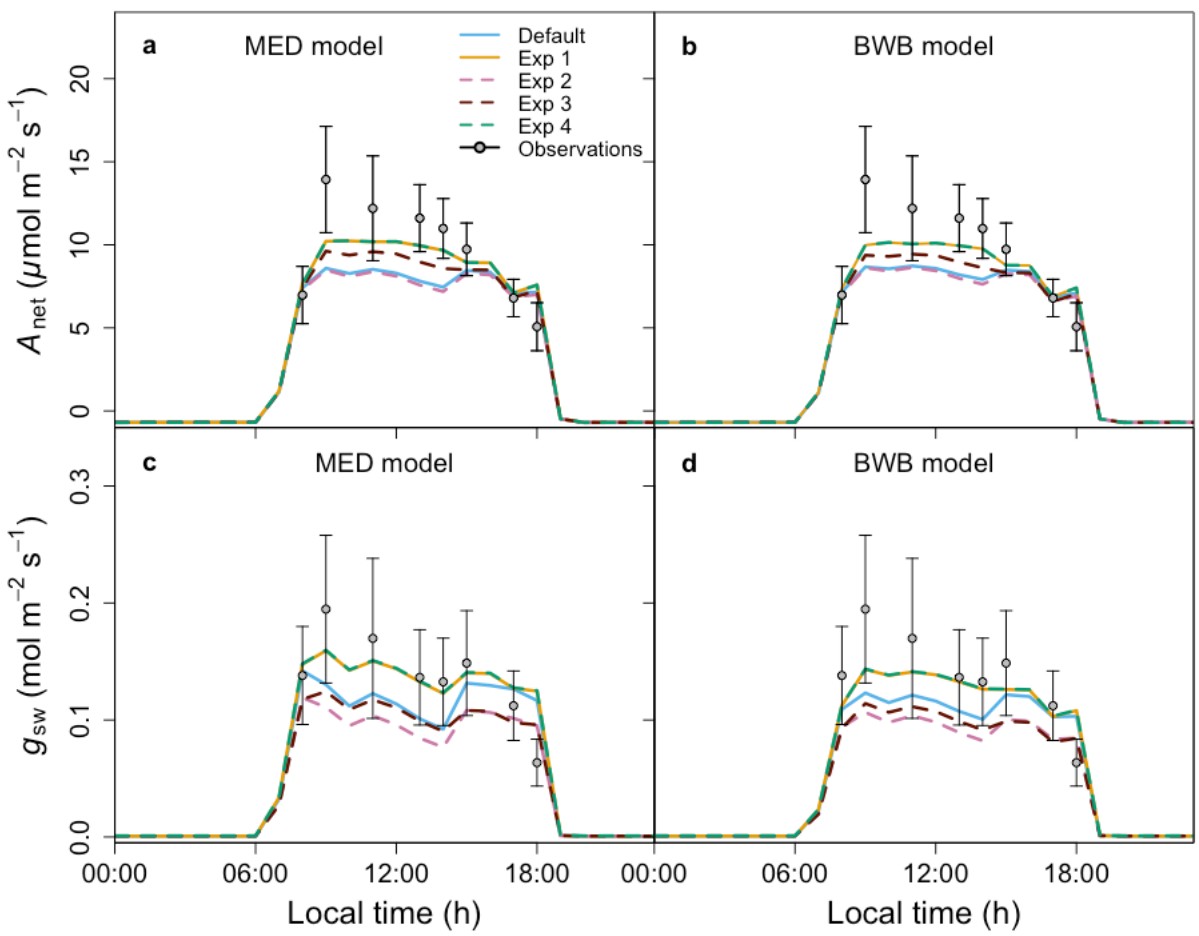

**Fig. 7. Comparison between the model outputs and measurements on 21 April 2016 for (a) the MED model net photosynthesis ($A_{net}$), (b) the BWB model $A_{net}$, (c) the MED model stomatal conductance ($g_{sw}$), and (d) the BWB model $g_{sw}$ with different soil water stress effects on parameters in FATES (Table 3).**

## 4 Discussion

### 4.1 Advances in understanding model difference

We implemented the MED stomatal model in FATES and compared model projections of $CO_2$ and water vapor exchange to the existing BWB formulation. The two models diverged considerably in the responses of both leaf-level ($g_{sw}$ and $A_{net}$, Fig. 1&2) and canopy-level (ET and GPP, Fig. 3&4) fluxes to a wide range of radiation, air temperature, $VPD_a$, and $CO_2$ concentration with default stomatal slope parameter ($g_1$). When parameterization of $g_1$ was harmonized between the MED and BWB formulations, the difference was much smaller in responses to varying radiation, temperature and $CO_2$ conditions but were markedly apparent at $VPD_a$ above 1.5 kPa.

Our analysis of the general model responses against synthetic climate forcing presents some advantages over previous evaluations. First, some studies found different stomatal conductance models varied considerably in water-limited regions (Knauer et al., 2015; Morales et al., 2005), but were unable to attribute the difference to specific climate

forcing as all factors, such as temperature and humidity are closely related (Galbraith et al., 2010; Rowland et al., 2015). In their recent experimental study of a tropical forest, Smith et al. (2020) found that stomatal response to $VPD_a$, rather than to $T_a$, is the primary mechanism for high-temperature photosynthetic declines in tropical forests by separating the temperature effect and $VPD_a$ effect. This observation, along with our findings highlighted the need to improve the representation of stomatal conductance response to $VPD_a$ in models. Second, most previous modeling studies relied on evaluating model performance against benchmarks such as eddy-covariance data and remote-sensing products (De Kauwe et al., 2015), which were limited to the current climate conditions and ecoregions. To test model behaviors under all possible climate change scenarios, our studies designed simulations driven by a wide range of climate forcing data. Third, understanding model response to synthetic climate forcing (Fig. 1-4) is a powerful diagnostic tool because the model outputs can be evaluated in comparison to known and measurable physiological responses to environmental variation, such as radiation and $CO_2$. The model outputs of GPP and ET also provide insight into how leaf-level responses influence the emergent ecosystem-scale responses, which is relevant for forecasting the responses of ecosystems and biomes to climate change. Fourth, by using the calibrated and default parameters to run the models, we were also able to separate effects of model structure (i.e., stomatal model choice) and parameterization (i.e., $g_0$ and $g_1$) on model differences.

As highlighted previously by Franks et al. (2017), the influence of parameterization dominated potential differences of $g_{sw}$ and ET due to model choice, further emphasizing the need to develop robust approaches to estimate $g_1$ and understand covariance with environmental drivers, such as soil moisture availability, and other leaf traits that may facilitate the use of trait-trait or trait-environment relationships to enable model parameterization (De Kauwe et al., 2015; Héroult et al., 2013; Lin et al., 2015; Wu et al., 2020). However, different $g_1$ values did not markedly change the magnitudes of $A_{net}$ and GPP, suggesting that the difference of $g_1$ propagates to the simulation of intercellular $CO_2$ first and finally to $A_{net}$ with attenuated effects. The structural difference is attributable to the different representation of humidity in the BWB and MED models (i.e., $H_s$ vs. $VPD_s$) and are consistent with the previous studies (Rogers et al., 2017a; Knauer et al., 2015; Franks et al., 2017). $g_{sw}$ simulated by the power function of the MED model decreases hyperbolically while that simulated by the linear function of the BWB model drops steeply. The nonlinear response of $g_{sw}$ to $VPD_a$ when using the MED model is supported by some observations (Marchin et al., 2016; Héroult et al., 2013; Wang et al., 2009; Domingues et al., 2014), but more measurements of leaf-level $VPD_a$ responses would be valuable. Our results suggest that when implemented in a dynamic vegetation demography model (FATES) the choice of stomatal model only has a small effect on projections of leaf and canopy level $CO_2$ and water vapor fluxes under conditions of $VPD_a$ below 1.5 kPa. Under higher $VPD_a$, higher $g_{sw}$ values were simulated using the MED model compared with the BWB model and led to higher $A_{net}$, ET, and GPP. This suggests that the MED formulation would predict tropical evergreen broadleaf forests to be more resistant to extreme atmospheric drought than with the BWB formulation. As the global surface temperature is projected to increase, the $VPD_a$ is also expected to increase (Ficklin and Novick, 2017; Yuan et al., 2019; Kolby Smith et al., 2016). Therefore, the difference between the two models under high $VPD_a$ conditions will lead to radically different ecosystem carbon and water dynamics under future climate change scenarios.

### 4.2 Model responses under simulated water stress

Our field campaign, which occurred during the 2016 ENSO event, enables us to evaluate model performance under various climate conditions, including extreme drought. Overall, simulations made with FATES with both $g_{sw}$ models captured the dynamics of measured upper canopy leaf-level fluxes well, confirming the utility of the current stomatal conductance models in LSMs for non-stressed conditions.

However, at the peak of the dry season this underestimation of $g_{sw}$ and $A_{net}$ was notable and resulted in part from application of a soil water stress ($\beta$) factor used to modify leaf physiology in response to reduced soil moisture content. In FATES, the $\beta$ factor affects $g_0$ and $V_{cmax}$ through an empirical modification. Experimental evidence about how physiological parameters change in response to soil water conditions is diverse. Some previous studies found $g_1$ was relatively stable under water stress (Gimeno et al., 2016; De La Motte et al., 2020; Xu and Baldocchi, 2003). Other studies found a range of responses of $g_1$ to drought across different plant species (Miner and Bauerle, 2017; Zhou et al., 2013). For $g_0$, it was reported to decrease under water stress (Miner and Bauerle, 2017; Misson et al., 2004), but also to show no response to drought (Barnard and Bauerle, 2013). Drought nearly universally lowered $V_{cmax}$ in plants (Zhou et al., 2013). However, some argued that effects of mild and moderate droughts on $V_{cmax}$ were negligible (Aranda et al., 2012; Bota et al., 2004; Cano et al., 2013), others showed a range of responses resulting in a 10 %-25 % reduction in $V_{cmax}$ (Galmés et al., 2007; Grassi and Magnani, 2005; Keenan et al., 2010; Limousin et al., 2010; Misson et al., 2010; Wilson et al., 2000; Zait and Schwartz, 2018).

Despite previous extensive experimental studies of the $\beta$ effects on plant physiological parameters, understanding of the results of applying $\beta$ effects in models is still inadequate. The uncertainty of the $\beta$ calculation is a major challenge. Based on the equations, the $\beta$ factor is a function of soil water content, modified by parameters related to plant response, root distribution, and soil properties. Due to the lack of in situ measurements, we only used general parameters for the $\beta$ factor in the simulations. Although soil moisture content was relatively well simulated (Fig. 6), root fraction and other soil properties were difficult to constrain due to scarce observations. In our study, by toggling on and off the $\beta$ effects on stomatal and photosynthetic parameters, we were able to learn more about how the calculation of $\beta$ influences model outputs. Overall, we found that the predictions of $g_{sw}$ and $A_{net}$ were closer to the measurements when the $\beta$ factor was treated as one (i.e., no stress). Similar studies also found that the implementation of the $\beta$ factor in CLM overestimated the drought-related productivity loss compared with the observations, biased the transpiration rate, or lacked diurnal variability (Powell et al., 2013; Kennedy et al., 2019; Bonan et al., 2014). To improve models, further systematic evaluation of the $\beta$ effects on photosynthetic capacity, stomatal conductance and mesophyll conductance in LSMs is highly recommended (Egea et al., 2011; Vidale et al., 2021). More mechanistic approaches such as representation of hydraulic limitations and chemical signaling through abscisic acid (ABA) are emerging as promising ways to represent the plant response to drought in LSMs but come with significant added complexity (Verhoef and Egea, 2014; Sperry and Love, 2015; Kennedy et al., 2019).

### 4.3 Implications for evaluating model performance

Our FATES sensitivity analysis used synthetic meteorological forcing to enable us to isolate the impacts of individual abiotic drivers on model behaviors. By adjusting the specific humidity concurrently with air temperature, we were able to isolate the model response to changing air temperature from typically concurrent change in $VPD_a$. We believe that our sensitivity analysis should be included as a routine approach for evaluating changes in model behavior during model development activities. Using similar simulations regularly during development would provide a powerful check on unexpected or unintended changes related to any changes in structure or parameters.

When the models are driven by synthetic climate forcing, special attention should be paid to the change of the environment conditions at the leaf surface, to which plants directly respond. In most LSMs, leaf surface temperature ($T_l$) is the balance of environmental drivers and leaf biophysical activities, and it is one of the most important variables regulating leaf biochemical responses such as photosynthesis and respiration (Kumarathunge et al., 2019; Leuning, 2002). But as $T_l$ is an emergent variable in FATES we could only control $T_a$ rather than $T_l$ in our sensitivity analysis simulations. In scenarios with changing radiation, although we have fixed $T_a$, $T_l$ was also increasing (Fig. S4a), which resulted in slight decreasing trends of $g_{sw}$ and $A_{net}$ in response to radiation as $T_l$ exceeded the temperature optimum of $g_{sw}$ and $A_{net}$ (Fig. 1d&2d). But the influence of $T_l$ change was limited for other response curves (Fig. S4).

Parameterization of $g_0$ has been shown critical for predicting ecosystem fluxes (De Kauwe et al., 2015; Barnard and Bauerle, 2013). However, there is little agreement on how to parameterize $g_0$ due to different definitions and measurement approaches for this parameter. Whether $g_0$ should be an intercept from data fitting, a minimum threshold when $A_{net}$ approaches zero, a night time $g_{sw}$, or the cuticular conductance is still an active research topic (Lombardozzi et al., 2017; Duursma et al., 2019; Lamour et al., 2022; Davidson et al., 2022). The slope parameter $g_1$ we used in the model was from Lin et al. (2015), estimated with the assumption that $g_0$ was zero. In our implementation, we not only included a non-zero $g_0$ in the numerical calculation of $g_{sw}$, but also set a small positive value for $g_0$ to prevent $g_{sw}$ becoming zero or negative when $A_{net}$ approaches zero. In this way, the leaf stomatal resistance (i.e., the reverse of $g_{sw}$) will not become infinitive during the simulations. To understand how different $g_0$ values influence $g_{sw}$ and $A_{net}$, we tested the sensitivity of $g_{sw}$, $A_{net}$, ET, and GPP to different $g_0$ values with our synthetic climate forcing listed in Table 2. In addition to the simulations with our default value (1000 $\mu$mol m$^{-2}$ s$^{-1}$), the $g_0$ was set zero or the commonly adopted value of 10000 $\mu$mol m$^{-2}$ s$^{-1}$ (Sellers et al. 1996). A comparison of $g_0$=0 and $g_0$=1000 $\mu$mol mol$^{-1}$ showed a very minor effect on the model response of $g_{sw}$. Using the ten-fold larger estimate for $g_0$ (10000 $\mu$mol mol$^{-1}$) only resulted in a small effect on the magnitudes of $g_{sw}$, $A_{net}$, ET, and GPP (Fig. S5-S8).

For the model evaluation against site-level measurements, we found it is necessary to check the consistency of climate forcing used in models and that measured by the instruments (Héroult et al., 2013). In our study, the in situ measured $T_a$ and $VPD_a$ were higher than those recorded by a nearby meteorological station (Fig. 5a-5d). The mismatch was partially related to the challenge of matching leaf chamber conditions with ambient conditions, avoiding condensation in the leaf chamber, or the use of a pump to move air across the leaf surface during gas exchange measurements. The

slight deviation of modeled $g_{sw}$ and $A_{net}$ against measurements when soil was relatively wet (measured in February, March, and May) can be partly attributed to the mismatch of $T_a$ and $VPD_a$ used in the model compared with the in situ
measurements. In this study we evaluated the inclusion of the MED model in FATES in a tropical forest, however future efforts could include evaluations at sites of different ecosystem types, and at regional and global scales for carbon and water cycles, particularly at sites where $VPD_a$ routinely rises above 1.5 kPa.

**5 Conclusions**

Implementing new plant physiological theories such as the optimal stomatal conductance model, into dynamic
vegetation models is crucial to keep the models up to date and to enable the exploration of new behaviors and capacities to understand potential ecosystem responses to global change. In this study, we added the optimality-based Medlyn model into the state-of-the-art dynamic vegetation model FATES as an alternative to the default Ball-Woodrow-Berry model and then tested model behaviors in response to key independent climate forcing. Our model evaluation demonstrated that the major difference between the two models was caused by the parameterization of the stomatal
slope parameter ($g_1$). When parameters were harmonized, the potential for markedly different projections of water vapor and $CO_2$ fluxes between stomatal conductance models only occurred as $VPD_a$ rose above 1.5 kPa. We also compared model performance with gas exchange measurements from an evergreen tropical forest. Modeled $CO_2$ and water vapor fluxes in the dry season of a drought year were similar between models and closely matched observations, except at the peak of the dry season when a soil moisture correction factor was used to adjust physiological parameters.
After removing this adjustment, projections for both models improved. Our study showed that the parameterization of $g_1$ and the application of the correction factor associated with decreasing soil moisture content are the key targets for improving model representation of $CO_2$ and water fluxes in tropical forests.

**Code availability**

The FATES model is available at https://github.com/NGEET/fates (https://doi.org/10.5281/zenodo.3825474, FATES
Development Team, 2020b). The specific FATES version used in this study is the one that merged the Medlyn model with git commit "9a4627a" and the version corresponds to tag "sci.1.37.0_api.11.2.0" (https://doi.org/10.5281/zenodo.5851984, FATES Development Team, 2020a). FATES was run here within CLM5. The latest release version of CLM5 is available at https://github.com/ESCOMP/ctsm (https://doi.org/10.5281/zenodo.3779821, CTSM Development Team, 2020), which is also the version used in this
study. Scripts to run all the model experiments, create synthetic climate forcing and analyze model outputs are available at https://github.com/Qianyuxuan/Scripts_for_papers/tree/main/Medlyn_model (https://doi.org/10.5281/zenodo.5854740, Li and Serbin, 2022).

## Author contributions

QL, AR, and SS designed the simulations and QL carried them out. QL, AR, SS, and JL analyzed the results. QL
prepared the manuscript with contributions from all co-authors.

## Competing interests

The authors declare that they have no conflict of interest.

## Acknowledgements

This work was supported by the Next-Generation Ecosystem Experiments in the Tropics (NGEE-Tropics) project
supported by the U.S. DOE, Office of Science, Office of Biological and Environmental Research, and through the
United States Department of Energy contract No. DE- SC0012704 to Brookhaven National Laboratory.

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
