# Peer review of "Implementation and evaluation of the unified stomatal optimization approach in the Functionally Assembled Terrestrial Ecosystem Simulator (FATES)"

_Geoscientific Model Development, 2021_

## Referee Comment (RC1)

**Implementation and evaluation if the unified stomatal optimization approach in the Functionally Assembled Terrestrial Ecosystem Simulator (FATES)**

In this paper Li et al., implement the optimality-based stomatal conductance model from Medlyn et al., (2011) (MED) into the FATES dynamic vegetation demography model. They compare the behaviour of this model to the existing empirical Ball-Woodrow-Berry (BWB) model. Firstly they assess how the response of simulated carbon and water fluxes to meteorological drivers ($CO_2$ concentration, air temperature, radiation, and VPD) differ between the two models to aid in understanding model representation and behaviour. These changes in meteorological drivers are applied in a sensitivity study, and differences arising from model parameterisation versus model structure are explored over a wide range on environmental conditions. Secondly, the authors evaluate the performance of each model at a tropical forest site in Panama. Thirdly the authors explore the application of the soil moisture stress function (the β factor) at different points in the physiological pathway which modifies simulated photosynthesis and stomatal conductance according to available soil moisture. This paper is well written and concise, and provides a nice evaluation of the impact of different representations of stomatal opening on carbon and water fluxes in a dynamic vegetation model, assessing the impact of both model structure and model parameterisation. I believe this paper fits within the scope of GMD and would be of interest to readers. I have a few comments below:

**Introduction:**
- Lines 70 to 80: Please add which models do what with regards to the β factor i.e. which models apply β to $g_0$ and/or $g_1$, and which models apply it to $V_{cmax}/J_{max}$, or elsewhere in the physiological pathway.
- The authors may find this paper relevant to their discussion on the application of the β factor within land surface models (particularly in the discussion around line 362): "On the Treatment of Soil Water Stress in GCM Simulations of Vegetation Physiology. 2021. Vidale et al., Frontiers in Environmental Science. https://doi.org/10.3389/fenvs.2021.689301".

**Methods:**
- Section 2.1: For clarity, can the authors be a bit more explicit about how they implemented the MED scheme, was it as straightforward as replacing equation 1 with equation 2 and adding the β factor?
- What is the photosynthesis scheme used to calculate $A$?
- Line 110: What measurements are being made at the site to compare with the simulations?
- Please add more detail about the model and simulations for clarity. FATES is initialised with real-world forest inventory data – so for these simulations that are at a single study site what does that represent – a single tree, an area of forest? Later on PFT specific parameterisations for the $g_1$ parameter are used, so are there different PFTs each with their own cohort structure? What meteorological forcing is required to drive the model, at what temporal resolution? Is the driving data provided by the test site, or from elsewhere? How is LAI modelled? A bit more clarity on the model and how it is run is required for those not familiar with FATES or CLM.

**Results:**
- What causes the difference between MED and BWB in VPD response when VPD > 1.5kPa? What do observations suggest is a more realistic response? Are there any observations from this site for the tropical trees to try and help pin down how $A$ and $gs$ are responding?
- Why is there a bigger difference (comparing MED-B and MED-default) in simulated $gs$ compared to $A$?
- Why does simulated ET increase with increasing VPD when $gs$ decreases?
- Line 255: Can you explain the abrupt changes better – I don't really see that MED is behaving that differently to BWB, and VPD rarely gets as low as 0.1 kPa.
- Could Figs 5, 6,7 and 8 be condensed? I wonder whether the diurnal cycles for the days without measurements are necessary? The months could then be plotted side by side for easier comparison (and on the same scale for the Met vars to make it easier to see how conditions change by month as the dry season progresses)?
- Are there any observations of soil moisture at the site? In April it seems that although the model is simulating reduced soil moisture availability which depresses $A$ and $gs$, the measured $A$ and $gs$ remain as high as in other months. Could it not be that the simulation of soil moisture stress itself in the model is not right and more of an issue than where the β factor is applied? How are soil parameters set in the model? Are these informed by the site-level information?

**Discussion:**

- Line 320 onwards: I am unclear on the third point. It says the response curves of *Anet* and *gs* are directly comparable to the leaf-level gas exchange measurements, but these data are not shown anywhere and do not seem to be used in this evaluation. These indeed would be invaluable to help determine which model behaviour is more realistic, for example to help pin down the VPD response which is largely where the two models seem to diverge. If they are available could they be included?

- Some discussion around the $g_0$ parameter would be interesting. Studies have shown that the $g_0$ term affects predictions of *gs* at all times, not just when *A* is close to zero, making predictions of plant water use very sensitive to this parameter. Is it right to have a minimum conductance when *Anet* is zero? What are the authors' justification for including the $g_0$ term in the MED formulation? Did the authors look at sensitivity of simulated *gs*/*Anet* to $g_0$?

---

## Author Comment (AC1)

**Response to the comments of Anonymous Referee #1 (RC1)**

We thank the reviewers and the editor for their suggestions and comments on the manuscript. We have considered all their comments and hope that the revised draft properly addresses their suggestions. Please find our point-by-point replies below (colored in blue). A revised manuscript with tracked changes will be uploaded. Line numbers in our text refer to the no-markup version.

In this paper Li et al., implement the optimality-based stomatal conductance model from Medlyn et al., (2011) (MED) into the FATES dynamic vegetation demography model. They compare the behaviour of this model to the existing empirical Ball-Woodrow-Berry (BWB) model. Firstly they assess how the response of simulated carbon and water fluxes to meteorological drivers ($CO_2$ concentration, air temperature, radiation, and VPD) differ between the two models to aid in understanding model representation and behaviour. These changes in meteorological drivers are applied in a sensitivity study, and differences arising from model parameterisation versus model structure are explored over a wide range on environmental conditions. Secondly, the authors evaluate the performance of each model at a tropical forest site in Panama. Thirdly the authors explore the application of the soil moisture stress function (the β factor) at different points in the physiological pathway which modifies simulated photosynthesis and stomatal conductance according to available soil moisture. This paper is well written and concise, and provides a nice evaluation of the impact of different representations of stomatal opening on carbon and water fluxes in a dynamic vegetation model, assessing the impact of both model structure and model parameterisation. I believe this paper fits within the scope of GMD and would be of interest to readers. I have a few comments below:

We thank the reviewer for carefully reviewing our submission, providing constructive suggestions, and acknowledging the importance of our work.

**Introduction:**

- Lines 70 to 80: Please add which models do what with regards to the β factor i.e. which models apply β to $g_0$ and/or $g_1$, and which models apply it to $V_{cmax}/J_{max}$, or elsewhere in the physiological pathway.

We have included the corresponding model names at Lines 77-85: "To mimic the drought effects on ecosystems, some models have included a soil water stress factor (often denoted as $\beta$) which is used to reduce the "base rate" of stomatal model parameters, either $g_0$ (e.g., CLM, Lawrence et al., 2019), $g_1$ (e.g., G'DAY, Comins and McMurtrie, 1993; O-CN, Zaehle and Friend, 2010; CABLE, De Kauwe et al., 2015), or both (e.g., ORCHIDEE, Guimberteau et al., 2018). In some cases, it is also used to lower the maximum carboxylation rate of Rubisco ($V_{cmax}$) (e.g., CLM; O-CN; SIBCASA, Schaefer et al., 2008), both $V_{cmax}$ and the maximum rate of electron transport ($J_{max}$) (e.g., G'DAY), or directly $A$ (e.g., JULES, Best et al., 2011; Clark et al., 2011). Reductions in $A$ will further reduce $g_{sw}$. Some models also consider the soil water stress on mesophyll conductance (e.g., SIBCASA; ORCHIDEE)".

- The authors may find this paper relevant to their discussion on the application of the β factor within land surface models (particularly in the discussion around line 362): "On the Treatment of Soil Water Stress in GCM Simulations of Vegetation Physiology. 2021. Vidale et al., Frontiers in Environmental Science. https://doi.org/10.3389/fenvs.2021.689301".

We thank the reviewer for the suggestion. It's indeed a very relevant paper. We have added the following sentence at Lines 410-412: "To improve models, further systematic evaluation of the $\beta$ effects on photosynthetic capacity, stomatal conductance and mesophyll conductance in LSMs is highly recommended (Egea et al., 2011; Vidale et al., 2021)".

**Methods:**

- Section 2.1: For clarity, can the authors be a bit more explicit about how they implemented the MED scheme, was it as straightforward as replacing equation 1 with equation 2 and adding the β factor?

We have added some additional description at Lines 114-118: "Leaf-level $g_{sw}$ is central to the water, $CO_2$ and energy cycles in forests. It not only controls the water and $CO_2$ exchange, but also modifies the energy balance and biochemical processes. Similarly, in FATES, the variable $g_{sw}$ is used to model several processes such as the heat and water transfer and photosynthesis. The calculation of this variable is therefore complex and uses both analytical and numerical solutions to couple the equations describing each process".

- What is the photosynthesis scheme used to calculate A?

We added the model description at Lines 104-106: "In FATES, leaf-level photosynthesis ($A$) in $C_3$ plants is based on the model of Farquhar et al. (1980) as modified by Collatz et al. (1991). $A$ is calculated as the minimum of RuBP carboxylase (Rubisco) limited rate and RuBP regeneration rate (i.e., the light-limited rate). Net photosynthesis rate ($A_{net}$) is the difference between $A$ and leaf respiration".

- Line 110: What measurements are being made at the site to compare with the simulations?

We modified this sentence at Lines 132-134 as: "The Smithsonian Tropical Research Institute canopy crane provides access to the top of the forest canopy and allows us to compare our simulations with previous measurements of stomatal conductance and net photosynthesis rate (Wu et al., 2020; Rogers et al., 2017)".

- Please add more detail about the model and simulations for clarity. FATES is initialised with real-world forest inventory data – so for these simulations that are at a single study site what does that represent – a single tree, an area of forest? Later on PFT specific parameterisations for the g1 parameter are used, so are there different PFTs each with their own cohort structure? What meteorological forcing is required to drive the model, at what temporal resolution? Is the driving data provided by the test site, or from elsewhere? How is LAI modeled? A bit more clarity on the model and how it is run is required for those not familiar with FATES or CLM.

The inventory data has information of tree size distribution for the whole forest area. For simplicity, in our FATES simulations we assumed that the site is populated entirely by the broadleaf evergreen tropical (BET) tree plant functional type. We parameterized $g_1$ based on measurements of eight different species which all belong to BET tree category. The FATES model is driven by half-hourly longwave radiation, shortwave radiation, air temperature, specific humidity, precipitation, surface pressure, wind speed, and atmospheric $CO_2$ concentration. The synthetic climate forcing for the sensitivity runs was created by ourself, while that for evaluation against in situ measurements was adopted from the meteorological station at the site (Faybishenko et al., 2019). The leaf area of each cohort is calculated from leaf biomass and specific leaf area. Leaf biomass is controlled by the processes of phenology, allocation and turnover. Specific leaf area is a PFT-specific parameter. We clarified those points in the revised manuscript at Lines 140-141, 143-145, 180-181, 158-159, 197-199, and 109-111.

**Results:**

- What causes the difference between MED and BWB in VPD response when VPD > 1.5kPa? What do observations suggest is a more realistic response? Are there any observations from this site for the tropical trees to try and help pin down how A and gs are responding?

The difference between MED and BWB in VPD response when VPD > 1.5kPa is caused by the different formulation of humidity in the two models. $g_{sw}$ simulated by the power function of MED model decreases hyperbolically while that simulated by the linear function of BWB model drops steeply. The nonlinear response of $g_{sw}$ to VPD when using MED model is supported by some observations (Marchin et al., 2016; Héroult et al., 2013; Wang et al., 2009; Domingues et al., 2014). Rising global temperature will raise the VPD but not the RH and therefore a formulation, such as the Medlyn model, may be better able to capture the response of vegetation to future global change (Way et al., 2011; Katul et al., 2009; Rogers et al., 2017b). It is very challenging to control the ambient temperature when studying the response of $g_{sw}$ to relative humidity and we did not have any observations from this site about the response to relative humidity.

We added some of the above discussion at Lines 368-372: "$g_{sw}$ simulated by the power function of the MED model decreases hyperbolically while that simulated by the linear function of the BWB model drops steeply. The nonlinear response of $g_{sw}$ to $VPD_a$ when using the MED model is supported by some observations (Marchin et al., 2016; Héroult et al., 2013; Wang et al., 2009; Domingues et al., 2014), but more measurements of leaf-level $VPD_a$ responses would be valuable".

- Why is there a bigger difference (comparing MED-B and MED-default) in simulated gs compared to A?

The difference in $g_1$ directly influences the magnitude of $g_{sw}$, causing the significant divergence in $g_{sw}$. However, it propagates to the simulation of intracellular $CO_2$ first and finally to $A$, which is the minimum root of quadratic equation of co-limitation rates from Rubisco limited and RuBP limited photosynthesis rates. Similar patterns are also discerned for ET and GPP, in which other factors such as VPD and leaf area index take effect, attenuating the difference caused by $g_1$.

We added the above discussion at Lines 364-366: "However, different $g_1$ values did not markedly change the magnitudes of $A_{net}$ and GPP, suggesting that the difference of $g_1$ propagates to the simulation of intercellular $CO_2$ first and finally to $A_{net}$ with attenuated effects".

- Why does simulated ET increase with increasing VPD when gs decreases?

ET can be approximately represented as $ET = g_{sw} \times VPD$, where $g_{sw}$ can be expressed as $\frac{f(A,CO_2)}{\sqrt{VPD}}$ for the Medlyn model. Therefore, ET is roughly proportional to $\sqrt{VPD}$. As a result, simulated ET increases with increasing $VPD$.

- Line 255: Can you explain the abrupt changes better – I don't really see that MED is behaving that differently to BWB, and VPD rarely gets as low as 0.1 kPa.

We agree that the difference is not clear. So we have deleted this sentence to prevent confusion in the revised manuscript.

- Could Figs 5, 6,7 and 8 be condensed? I wonder whether the diurnal cycles for the days without measurements are necessary? The months could then be plotted side by side for easier comparison (and on the same scale for the Met vars to make it easier to see how conditions change by month as the dry season progresses)?

We agree that the modeling results for the days without measurements are not very necessary for the evaluation purpose. We also adopted the suggestion to plot all evaluation results together into Fig. 5. For the dry day in April, the climate drivers did not exhibit distinct trends compared with other months, but the soil moisture content was quite low (see Fig. 6). To be consistent, we also modified Fig. 7 to only include results with corresponding measurements.

[Figure]

**Fig. 5. (a-d)** Diurnal change in climate forcing, **(e-h)** model-data comparison of net photosynthesis rate ($A_{net}$), **(i-l)** model-data comparison of stomatal conductance ($g_{sw}$) for four field campaign dates. In panel (a-d), lines and filled points represent climate forcing data used in FATES and in situ measurements, respectively. Different colors are for different types, black for $T_a$, red for PAR, blue for $VPD_a$, green for atmospheric $CO_2$ concentration, and gold for $H_a$. In panel (e-l) shading areas represent range of FATES model ensemble results with different measured $g_1$ values for different species, while lines represent the averages of these ensemble results. Blue shading areas and lines are for results from the MED model, and red for the BWB model. Gray filled circles for the measured data represent averages across species. Black error bars for the measured data represent the 95 % CI across species. Columns correspond to days of measurements and are presented in chronological order for 17 February, 10 March, 21 April, and 24 May.

[Figure]

**Fig. 7. Comparison between the model outputs and measurements on 21 April 2016 for (a) the MED model net photosynthesis ($A_{net}$), (b) the BWB model $A_{net}$, (c) the MED model stomatal conductance ($g_{sw}$), and (d) the BWB model $g_{sw}$ with different soil water stress effects on parameters in FATES (Table 3).**

- Are there any observations of soil moisture at the site? In April it seems that although the model is simulating reduced soil moisture availability which depresses A and gs, the measured A and gs remain as high as in other months. Could it not be that the simulation of soil moisture stress itself in the model is not right and more of an issue than where the β factor is applied? How are soil parameters set in the model? Are these informed by the site-level information?

We thank the reviewer for prompting us to explore this further and feel the addition of the new figures and discussion markedly improve the manuscript. Soil water stress is a function of soil water content and parameters related to plant response, root distribution and soil properties. We added information about the stress factor at Lines 123-127: "The soil wilting factor is a bounded linear function of soil matric potential, defined by two parameters, the soil water potential at (and above) which stomata are fully open, and the value at which stomata are fully closed. The soil matric potential is related to the soil water content, soil texture, and organic matter content. The root fraction is determined by PFT-specific root distribution parameters".

Due to the lack of in situ parameters, we only used general soil and root parameters in the simulations. We compared the modeled soil water contents against soil moisture products of ECMWF Reanalysis data version 5 (ERA5) (Hersbach et al., 2018) for the site. Results show that the model captured the dynamics and approximate magnitude of the average soil water content (Fig. 6). Similar patterns were observed for different layers throughout the first three meters of the soil column (Fig. S3). Although soil moisture was relatively well simulated, root fraction and other soil properties were difficult to constrain due to scarce observations. The relatively large mismatch of modeled and measured $A_{net}$ and $g_{sw}$ in April compared with other months is likely to be related to soil water stress as we have ruled out the influence from other climate factors (Fig. 5). Our results indicate that the simulations that treated the stress factor as one (i.e., no stress) for all parameters produced higher $A_{net}$ and $g_{sw}$ and matched the observations best. This suggests the photosynthesis and stomatal conductance in tropical forests are more resilient to drought than are currently represented by FATES.

[Figure]

**Fig.6. Annual cycle of modeled volumetric soil water content (blue line) and corresponding soil water stress (β) factor (green line) from FATES simulation, and ERA5 reanalysis soil water content data (orange line) at the San Lorenzo field site in 2016. The soil water content data are means across all soil layers. For the β factor, "1" represents fully saturated soil, while "0" represents very dry soil. Vertical gray lines indicate the four campaign dates in 2016.**

[Figure]

**Fig. S3. Annual cycle of modeled soil water content (blue lines) and corresponding ERA5 reanalysis data (orange lines) for different soil layers.**

We added the first figure above to the revised manuscript (Fig. 6) and the second to the supplemental material (Fig. S3). Besides, we added some texts in the Methods at Lines 205-208 as:

"However, whether the calculation of the $\beta$ factor can truly reflect soil water conditions is unclear. To the best of our knowledge, the relevance of the $\beta$ factor has not been rigorously tested for tropical ecosystems, in comparison with measured $g_{sw}$ and $A_{net}$, either. We therefore first compared the modeled soil water content and $\beta$ factor against soil moisture products of ECMWF Reanalysis data version 5 (ERA5) (Hersbach et al., 2018)".

Corresponding results were added at Lines 305-309 as:

"Compared with the ERA5 soil moisture products, FATES generally captured the magnitude and trend of the observed average soil water content at the San Lorenzo site (Fig. 6). FATES also simulated the soil water content well for different layers of the soil column (Fig. S3). By April 2016, at the peak of the dry season in a dry year, the simulated soil moisture stress factor (averaged over all the soil layers) reached an annual minimum (0.7), corresponding to the observed soil moisture drying trend (Fig. 6)".

We also added some discussion at Lines 400-410 as:

"Despite previous extensive experimental studies of the $\beta$ effects on plant physiological parameters, understanding of the results of applying $\beta$ effects in models is still inadequate. The uncertainty of the $\beta$ calculation is a major challenge. Based on the equations, the $\beta$ factor is a function of soil water content, modified by parameters related to plant response, root distribution, and soil properties. Due to the lack of in situ measurements, we only used general parameters for the $\beta$ factor in the simulations. Although soil moisture content was relatively well simulated (Fig. 6), root fraction and other soil properties were difficult to constrain due to scarce observations. In our study, by toggling on and off the $\beta$ effects on stomatal and photosynthetic parameters, we were able to learn more about how the calculation of $\beta$ influences model outputs. Overall, we found that the predictions of $g_{sw}$ and $A_{net}$ were closer to the measurements when the $\beta$ factor was treated as one (i.e., no stress). Similar studies also found that the implementation of the $\beta$ factor in CLM overestimated the drought-related productivity loss compared with the observations, biased the transpiration rate, or lacked diurnal variability (Powell et al., 2013; Kennedy et al., 2019; Bonan et al., 2014)".

**Discussion:**

- Line 320 onwards: I am unclear on the third point. It says the response curves of Anet and gs are directly comparable to the leaf-level gas exchange measurements, but these data are not shown anywhere and do not seem to be used in this evaluation. These indeed would be invaluable to help determine which model behaviour is more realistic, for example to help pin down the VPD response which is largely where the two models seem to diverge. If they are available could they be included?

We agree that such data would be really useful, but we did not measure the corresponding leaf-level response curves. The value of this synthetic approach is being able to observe how the model simulates theoretical responses to environmental variables that are well understood by physiologists and ensure that expected behaviors are reproduced e.g., a temperature optimum. The focus in this study was to understand model differences in terms of model response to key climate forcing (as at Lines 97-99).

We made this point clearer in the revised manuscript at Lines 352-354: "Third, understanding model response to synthetic climate forcing (Fig. 1-4) is a powerful diagnostic tool because the model outputs can be evaluated in comparison to known and measurable physiological responses to environmental variation, such as radiation and $CO_2$".

- Some discussion around the $g_0$ parameter would be interesting. Studies have shown that the $g_0$ term affects predictions of gs at all times, not just when A is close to zero, making predictions of plant water use very sensitive to this parameter. Is it right to have a minimum conductance when Anet is zero? What are the authors' justification for including the $g_0$ term in the MED formulation? Did the authors look at sensitivity of simulated gs/Anet to $g_0$?

We appreciate the reviewer's comments and agree that the value of $g_0$ should be given attention. There is less consensus for the parameterization of $g_0$ due to different definitions and measurements of this parameter. Whether $g_0$ should be an intercept from data fitting, a minimum threshold when $A_{net}$ approaches zero, a night time $g_{sw}$, or the cuticular conductance is still an active research topic (Lombardozzi et al., 2017; Duursma et al., 2019; Lamour et al., 2022). The slope parameter $g_1$ we used in the model was from Lin et al. (2015), estimated with the assumption that $g_0$ was zero. In our implementation, we not only included a non-zero $g_0$ in the numerical calculation of $g_{sw}$, but also set a small positive value for $g_0$ to prevent $g_{sw}$ to become zero or negative when $A_{net}$ approaches zero or negative. In this way, the leaf stomatal resistance (i.e., the reverse of $g_{sw}$) will not become infinitive during the simulations. Besides, including a user-defined $g_0$ in the equation will also encourage further exploration about the different usage of $g_0$. For example, some studies find $g_0$ was related to soil water condition (Misson et al., 2004) or heatwave (Duarte et al., 2016).

To address the reviewer's comments further, we tested the sensitivity of $g_{sw}$, $A_{net}$, ET and GPP to different $g_0$ with our synthetic climate forcing. In addition to the simulations with our default value (1000 $\mu$mol m$^{-2}$ s$^{-1}$), the $g_0$ was set zero or the commonly adopted value of 10000 $\mu$mol m$^{-2}$ s$^{-1}$ (Sellers et al. 1996). Results showed that compared with a zero $g_0$, the default $g_0$ had almost no influence on the model response of $g_{sw}$. Using the ten-fold larger estimate for $g_0$ (10000 $\mu$mol mol$^{-1}$) only resulted in a small effect on the magnitudes of $g_{sw}$, $A_{net}$, ET, and GPP (Fig. S5-S8).

[Figure]

**Fig. S5. The responses of stomatal conductance ($g_{sw}$) to scenarios (a) Radiation, (b) $CO_2$, (c) $VPD_a$, and (d) $T_a$ for the three MED model setups with different $g_0$ values. $g_0$ is in $\mu$mol m$^{-2}$ s$^{-1}$.**

[Figure]

Fig. S6. The responses of net photosynthesis ($A_{net}$) to scenarios (a) Radiation, (b) $CO_2$, (c) $VPD_a$, and (d) $T_a$ for the three MED model setups with different $g_0$ values. $g_0$ is in $\mu$mol m$^{-2}$ s$^{-1}$.

[Figure]

**Fig. S7. The responses of evapotranspiration (ET) to scenarios (a) Radiation, (b) CO₂, (c) *VPD*ₐ, and (d) *T*ₐ for the three MED model setups with different *g*₀ values. *g*₀ is in *μ*mol m⁻² s⁻¹.**

[Figure]

**Fig. S8. The gross primary productivity (GPP) to scenarios (a) Radiation, (b) CO₂, (c) *VPD*ₐ, and (d) *T*ₐ for the three MED model setups with different $g_0$ values. $g_0$ is in $\mu$mol m⁻² s⁻¹.**

We have added the above figures to the supplemental material (Fig. S5-S8) and one paragraph at Lines 433-446 as: "Parameterization of $g_0$ has been shown critical for predicting ecosystem fluxes (De Kauwe et al., 2015; Barnard and Bauerle, 2013). However, there is little agreement on how to parameterize $g_0$ due to different definitions and measurement approaches for this parameter. Whether $g_0$ should be an intercept from data fitting, a minimum threshold when $A_{net}$ approaches zero, a night time $g_{sw}$, or the cuticular conductance is still an active research topic (Lombardozzi et al., 2017; Duursma et al., 2019; Lamour et al., 2022; Davidson et al., 2022). The slope parameter $g_1$ we used in the model was from Lin et al. (2015), estimated with the assumption that $g_0$ was zero. In our implementation, we not only included a non-zero $g_0$ in the numerical calculation of $g_{sw}$, but also set a small positive value for $g_0$ to prevent $g_{sw}$ becoming zero or negative when $A_{net}$ approaches zero. In this way, the leaf stomatal resistance (i.e., the reverse of $g_{sw}$) will not become infinitive during the simulations. To understand how different $g_0$ values influence $g_{sw}$ and $A_{net}$, we tested the sensitivity of $g_{sw}$, $A_{net}$, ET, and GPP to different $g_0$ values with our synthetic climate forcing listed in Table 2. In addition to the simulations with our default value (1000 $\mu$mol m⁻² s⁻¹), the $g_0$ was set zero or the commonly adopted value of 10000 $\mu$mol m⁻² s⁻¹ (Sellers et al. 1996). A comparison of $g_0$=0 and $g_0$=1000 $\mu$mol mol⁻¹ showed a very minor effect on the model response of $g_{sw}$. Using the ten-fold larger estimate for $g_0$ (10000 $\mu$mol mol⁻¹) only resulted in a small effect on the magnitudes of $g_{sw}$, $A_{net}$, ET, and GPP (Fig. S5-S8)".

---

## Author Comment (AC2)

**Response to the comments of Anonymous Referee #2 (RC2)**

We thank the reviewers and the editor for their suggestions and comments on the manuscript. We have considered all their comments and hope that the revised draft properly addresses their suggestions. Please find our point-by-point replies below (colored in blue). A revised manuscript with tracked changes will be uploaded. Line numbers in our texts refer to the no-markup version.

-This paper, submitted to the journal, Geoscientific Model Development (GMD), by Q. Li, Serbin, Lamour, Davidson, Ely, and Rogers, entitled "Implementation and evaluation of the unified stomatal optimization approach in the Functionally Assembled Terrestrial Ecosystem Simulator (FATES)", is a well-written paper that could be accepted after mild revisions. The topic is important for understanding climate and land-surface processes better, and the modeling exhibited here is first rate. I detail my comments below.

I am impressed by the fact that the authors have started the FATES model runs with real-world forest inventory data, as stated on Line 115.

We appreciate the reviewer taking the time to review and providing valuable feedbacks to this study. We are very excited to contribute to the development and evaluation of FATES.

-Line 136: "we set the precipitation to $1.47\hat{}10\text{-}5$ mm/s" = 1mm/day? So it is always raining? Is this consistent with the humidity or VPD values of the model experiments? Is it consistent with the PAR values of the model experiments?

We have varied precipitation but did not find it was related to the change of humidity in the model. It is the specific humidity that determines the humidity and influences VPD. In our sensitivity runs with synthetic climate forcing, precipitation does not need to be consistent with humidity or PAR. The value $1.47\times10^{-5}$ mm/s (i.e., 1.3 mm/day) was calculated as the annual averaged precipitation and used for the standard condition when we explored model responses to other climate variables. Because our sensitivity runs were only conducted on a short time scale, the precipitation was not able to influence soil moisture. We changed the unit of this value to "mm/day" at Line 165 to make it more concise.

-Fig 5.: There is not much difference in A_net or g_sw for the 3 days for either BWB_mean or MED_mean, even though the average-peak PAR increases from 700 to 1200 to 1500 mol/m2/s for the 3 successive days. This approximate independence of g_sw on PAR is what can be expected from Fig. 1a, for PAR > 500 mol/m2/s. But from Fig 1a, it might be expected that MED−default_mean and BWB_mean would differ by a factor of 2 in Fig. 5a. Is this Figure 5 actually for MED-B_mean instead of being for MED-default_mean? If it is, then the lack of difference between the modeled values for A_net or g_sw would make more sense.

We appreciate the reviewer's idea of linking the evaluation results with previous sensitivity analysis. Yes, the slope parameter $g_1$ in MED_mean and BWB_mean in Fig. 5 is calibrated based on the same field measurements, therefore equivalent to MED-B and BWB in Fig. 1. The reason why we did not name the MED simulation as "MED-B" in Fig. 5 is that we tried to differentiate the parameterization for the evaluation simulations from that for the sensitivity simulations. To increase the connection between the figures, we added the following sentences at Lines 189-190:" Because $g_1$ was estimated for BWB and MED models based on the same measurements, $g_1$ was equivalent for the two models and the simulations resemble MED-B and BWB in section 2.3".

-Or should we be comparing to the ecosystem dependence shown in Fig. 2a, which shows little difference? I would expect the LICOR measurements to be done on a single leaf, instead of measuring over a larger ecosystem.

Yes, the LICOR measurements were conducted on a single leaf.

-The case of PAR < 500 mol/m2/s seems to be handled robustly for the date of May 25, in Fig. 8, where both BWB_mean and MED_mean are lower than the previous 2 days in May, particularly later in the afternoon on May

25. In this case of May 25, BWB_mean does seem to be 2 times higher than MED_mean, even in the morning, which might make a bit more sense if is for MED-default_mean instead of MED-B_mean, this time. On May 23 and on May 24, BWB_mean is 50% greater than MED_mean in the morning, but by mid-day, the models don't differ much. Maybe the higher VPD that is reached by mid-day on May 23 and May 24 effectively closes the pores, causing the models not to differ? May seems different than (the dry season of) February - April, in that VPD is 0 kPa at night for May.

We guess that the reviewer intended to express that MED_mean is two times higher than BWB_mean on May 25, and MED_mean is 50% greater than BWB_mean in the morning of May 23 and 24 based on previous Fig. 8c. As we stated above, we used the comparable $g_1$ parameters for BWB and MED models in the evaluation. The reason for higher MED_mean under those conditions could be that the modeled VPD at the leaf surface ($VPD_s$) is very low due to the low VPD in the air ($VPD_a$). See Fig. 5c in Franks et al. (2017), the MED model predicts markedly higher $g_{sw}$ than the BWB model when $VPD_s$ is very low.

-Lines 376-378: "Our method in keeping VPD in the air constant when studying model response to varying T_air by adjusting specific humidity concurrently is inspiring for other modelers." Such future inspiration of other modelers may indeed happen, but the language in this sentence is a bit presumptuous.

We modified this sentence to: "By adjusting the specific humidity concurrently with air temperature, we were able to isolate the model response to changing air temperature from typically concurrent change in $VPD_a$" at Lines 418-419.

-Line 619: citation for Pachauri et al. should have 51 authors instead of 10 authors.

We corrected this reference in the revised manuscript.

-Fig S2b: The r^2 value for the MED model is quite a bit lower than for the BWB model. Is this a real effect? Maybe the fit can be improved by removing a single outlier for MED at a value of Modeled g_sw = 0.24? It's ok sometimes to remove outliers when doing fits. And that outlier seems unusual, too, since it is a MED point that doesn't have a corresponding nearby BWB point like most of the other points do.

In that figure we combined results from the four campaigns to indicate the model overall performance in capturing the means of observations across all the species. The lower $R^2$ for $g_{sw}$ fitting when using the MED compared with the BWB is mainly contributed by the results in May, when MED model overestimated mean $g_{sw}$ whereas BWB model captured mean $g_{sw}$ relatively well. However, given the small number of measurements and large uncertainty range of both measurements and model results, we could not tell which model is superior simply based on the fitting of mean responses. We believe the evaluation would be more informative as more observations are available in the future.

**References**

Franks, P. J., Berry, J. A., Lombardozzi, D. L., and Bonan, G. B.: Stomatal function across temporal and spatial scales: Deep-time trends, land-atmosphere coupling and global models, Plant Physiol., 174, 583–602, https://doi.org/10.1104/pp.17.00287, 2017.